

# Metagenomics-based analysis of mobile genetic elements and antibiotic/metal resistance genes carried by treated wastewater

Chahnez Naccache[1], Chourouk Ibrahim[2,3], Abdennaceur Hassen[3] and Maha Mezghani Khemakhem[1]

[1] Laboratory of Biochemistry and Biotechnology (LR01ES05), Faculty of Sciences of Tunis, University of Tunis El Manar, Tunis El Manar, Tunisia
[2] Medical Biology Laboratories Unit ULB, Ministry of Health, Tunis, Tunisia
[3] Laboratory of Treatment and 11 Valorization of Water Rejects (LTVRH), Center of Research and Water Technologies (CERTE), Borj Cédria, Tunisia

Corresponding author
Chahnez Naccache,
chahneznaccache@yahoo.com

## ABSTRACT

Wastewater treatment plants in Tunisia are recognized as key locations for the spread of antibiotic and heavy metal resistance genes among bacteria. Despite the widespread presence of pollutants in these treatment systems, there is still a significant gap in our understanding of resistance dynamics. This study focused on analyzing the bacterial community and resistome-mobilome profiles of the Charguia wastewater treatment plant (WWTP). Using metagenomics sequencing, six samples from the influent, sludge, and effluent were thoroughly examined. Our research findings indicated the prevalence of *Proteobacteria* and high levels of *Bacteroidota*, *Firmicutes*, *Campylobacterota*, and *Patescibacteria*. After conducting a species level analysis, we identified important species such as *Pseudomonas psychrophila*, *Pseudomonas fragi*, *Pseudomonas lundensis*, *Acinetobacter johnsonii*, and *Thiothrix unzii* linked to antibiotic resistant genes (ARGs) like *mdtA* and *merR1* and heavy metal resistance genes (MRGs), including *czcA* and *cnrA*. Our study illustrated the persistence of specific species in the effluent due to the co-occurrence of ARGs/MRGs and mobile genetic elements (MGE). Notably, *IncQ* and *IncP* were found to be associated with *mdtA, mexR, arsR1*, and *merR*. The conclusions drawn from our research suggest that the WWTP has been potentially effective in reducing multidrug resistance.

# INTRODUCTION

Antimicrobial resistances (AMRs) pose a major threat to the environment and human health worldwide. Recent data from 2019 showed that antimicrobial bacteria (ARBs) were responsible for 1.27 million deaths (*Murray et al., 2022*). If left unchecked, the uncontrolled spread of antibiotic resistance could lead to a staggering 10 million deaths by 2050, exceeding the estimated number of deaths caused by cancer (*Aljeldah, 2022*). This global trend is mirrored in Africa, where the growing burden of AMR is particularly alarming

due to limited surveillance and control measures. Africa is particularly vulnerable, with 88 combinations of drug-resistant pathogens documented in the World Health Organization (WHO) region (*Mestrovic et al., 2022*). Excessive and inappropriate use of antibiotics in various fields, including veterinary, medicine, and agriculture, has led to the emergence of resistant microbial strains and impaired our ability to effectively combat certain infections.

Tunisia represents a specific case that highlights the consequences of widespread antibiotic use at the national level. In 2015, Tunisia was reported to be the second-largest consumer of antibiotics worldwide (*Klein et al., 2018*), underscoring the scale of antimicrobial consumption in the country. Although data from other African countries also point to rising resistance levels, Tunisia's high antibiotic use and increasing evidence of resistance in environmental samples suggest a particularly urgent need for local surveillance and mitigation strategies.

One of the primary environmental pathways for the spread of resistance is through aquatic systems, which serve as reservoirs for resistant bacteria and genes.

Aquatic environments such as rivers, groundwater, fish farms, and wastewater treatment systems have been identified as significant antibiotic reservoirs, as *Marti, Variatza & Balcazar (2014)* emphasized. In particular, wastewater treatment plants are considered a major source of various pollutants, including antibiotics, heavy metals, and biocides (*Chukwu et al., 2023*). These pollutants originate from urban, industrial, and pharmaceutical discharges and exert selective pressure on microbial communities, thereby promoting the survival of resistant bacteria and the dissemination of resistance genes (*Sambaza & Naicker, 2023*). This occurrence may lead to an increased prevalence of resistant bacteria in wastewater that evades conventional treatment methods (*Niestepski et al., 2020*).

While the global impact of wastewater on AMR is widely recognized, several studies have also explored this issue in the Tunisian context. Research efforts in Tunisia have shed light on the complex problem of antibiotic and heavy metal resistance in the country's environment. Significant progress has been made in understanding the prevalence of antibiotic resistance, particularly in wastewater and other environmental samples. Studies on sewage treatment plants, water samples, rivers, and seawater have provided valuable information about the antibiotics found. For example, aminoglycoside and phenicol antibiotics have been detected in seawater samples from Charguia and Chotrana as well as sewage treatment plants. In this context, Charguia, located on the outskirts of Tunis, is a highly urbanized and industrialized zone that receives a mix of domestic, hospital, and pharmaceutical wastewater, making it a hotspot for diverse pollutants and resistance genes. On the other hand, Chotrana is a semi-urban residential area where environmental monitoring has revealed the presence of antibiotics in coastal waters, despite its lower industrial activity. Another study found the common presence of sulfonamides, macrolides, and quinolones in sewage treatment plants in Tunis, Bizerte, Sousse, and Monastir. Notably, these regions are major coastal cities with high population densities and significant urban and touristic activities, contributing to complex wastewater profiles that impact local ecosystems.

Beyond the detection of antibiotics, numerous investigations have also identified multidrug-resistant bacteria in diverse environmental settings. *Bouamama, Bour & El (2006)* began this investigation by examining multidrug-resistant isolates in the marine species *Mytilus galloprovincialis* from the Bizerte Lagoon and discovered *Aeromonas hydrophila, Enterobacter sakazakii,* and *Pseudomonas cepacia.* In Tunisia's wastewater treatment plants, *Chouchani, Marrakchi & El Salabi (2011)* reported that beta-lactam resistance genes are most common in gram-negative bacteria, while *Rafraf et al. (2016)* found quinolone and beta-lactam resistance genes in Monastir. These genes have also been detected in pharmaceutical wastewater in northern Tunisia (*Tahrani et al., 2015*; *Tahrani et al., 2017*).

*Freitas et al. (2017)* reported linezolid-resistant *Enterococcus faecalis* strains in urban wastewater treatment plants. These resistant strains were also detected in clinical samples in Tunisia (*Raddaoui et al., 2020*) and in food-producing animals (*Elghaieb et al., 2019*). In addition, a study by *Ben Said et al. (2017)* investigated multidrug-resistant bacteria in sewage treatment plants and identified *Staphylococcus aureus* strains in the Charguia and El Menzeh 1 facilities. These strains showed resistance to penicillin, erythromycin, tetracycline, and clindamycin. *Hassen et al. (2020)* found multidrug-resistant *Klebsiella pneumoniae* strains in the treated wastewater of the El Menzeh 1 pilot plant and in streams of the Oued Rouriche Wadi in the city of Tunis.

To provide broader context, studies from other parts of the world have similarly demonstrated the presence of multidrug-resistant bacteria in wastewater environments. Research conducted in Brazil has uncovered resistant strains in wastewater, including *Aeromonas, Arcobacter, Stenotrophomonas, Streptococcus, Bacillus,* and *Shigella* (*Leroy-Freitas et al., 2022*; *Machado et al., 2023*). Other studies, conducted in Spain, Egypt, and China, have identified *Enterococcus spp., Staphylococcus aureus* and *Escherichia coli* in wastewater before they enter the environment (*López et al., 2019*; *Mehanni et al., 2023*; *Yuan, Guo & Yang, 2015*). These bacterial strains exhibit multidrug resistance and exhibit different levels of resistance to antibiotics such as tetracycline, ampicillin, amoxicillin, chloramphenicol, and erythromycin (*Mehanni et al., 2023*). *Moslah et al. (2018)* documented increased concentrations of these resistant strains in Tunisia, particularly in Charguia, Chotrana, and Rades.

In addition to antibiotic pressure, co-selection mechanisms, particularly those involving heavy metals, are increasingly recognized as key drivers of resistance. Heavy metals, present in the environment through agricultural and industrial discharges, can exert selective pressure similar to antibiotics (*Gupta, Sreekrishnan & Ahammad, 2022*). This influence is observed not only in rivers but also in wastewater treatment plants, as shown by *Di Cesare et al. (2016)* and *Gupta et al. (2023)*. In Tunisia, *Miloud et al. (2021)* examined this interaction in Bizerte, Rades, Gafsa, and Gabès and identified resistance genes for silver, copper, and mercury. Their results showed that 97.43% of the isolates had simultaneous resistance to both antibiotics and heavy metals. In addition, Bizerte Lagoon sediments have been found to be contaminated with heavy metals such as nickel, copper, zinc, lead, chromium, and cadmium, according to studies by *Ben Said et al.. (2010)* and *Zaaboub et al.*

*(2015)*. This comprehensive review highlights the complex relationship between antibiotic resistance and heavy metal contamination in different environments.

The transfer of resistance traits is not only influenced by selective pressures but also facilitated by mobile genetic elements (MGEs), which play a central role in horizontal gene transfer within wastewater ecosystems. Activated sludge is considered a key point for the transfer of both antimicrobial and heavy metal resistance genes (*Gibson et al., 2023*). Recent studies have focused on MGE such as plasmids, transposons, and integrons to gain a better understanding of how resistance genes are transferred horizontally between bacteria through processes such as transformation or conjugation, which has been extensively studied in the existing literature (*Li et al., 2021*; *Lood, Ertürk Bergdahl & Mattiasson, 2017*). Furthermore, there is growing interest in the involvement of bacteriophages that help transfer resistance genes through transduction (*Lood, Ertürk Bergdahl & Mattiasson, 2017*). In the sewage treatment plants of Tunis and El Menzeh, a study revealed class 1 integrons in isolates carrying aminoglycoside and trimethoprim resistance genes (*Hassen et al., 2020*). An evaluation of the Monastir wastewater treatment plant also revealed the presence of antimicrobial resistance genes (ARGs) and the IntI1 gene (class 1 integon integrase gene) (*Rafraf et al., 2016*). As for the plasmids, the IncP and IncFIB predominant in bacterial isolates from the Tunis and El Menzeh wastewater treatment plants were associated with ARGs-bearing *Enterobacteriaceae*, as reported by *Hassen et al. (2020)*. This collaborative research highlights the important role of activated sludge in the complex dynamics of resistance gene transmission and provides insights into specific discoveries in various wastewater treatment plants.

However, despite these important findings, significant knowledge gaps remain, particularly concerning the interaction between ARGs, MRGs, and MGEs in Tunisian wastewater systems. The need for continued research to understand the complex interactions between antibiotic resistance and heavy metal contamination underscores the importance of solving this problem. Tunisia has played an important role in the global discussion on antibiotic resistance, but further advances in research and the introduction of sustainable management strategies are important to effectively address this ever-evolving challenge.

Despite the increasing number of publications on ARBs and ARGs from wastewater treatment plants worldwide, the complex relationship between the core components of ARBs, ARGs/MRGs and MGE in the studied wastewater treatment plants in Tunisia remains poorly understood. Therefore, this study applies a metagenomic resistome-mobilome analysis to assess the occurrence and dynamics of resistance genes and mobile elements in the Charguia wastewater treatment plant, a representative biological treatment facility in Tunisia.

This strategy aims to address challenges by (i) conducting comprehensive screening of bacterial diversity and identification of resistant bacteria, (ii) profiling ARGs/MRGs and MGE, and (iii) studying the dynamics of the bacterial ecosystem within the activated sludge is analyzed acts as a transmission center of resistance and its removal efficiency.

## MATERIALS AND METHODS

### Wastewater sample collection and DNA extraction

In November, wastewater samples of 1L each were collected from three different lines within the Charguia sewage treatment plant over a period of two years, specifically 2022 and 2023. These lines were designated as INF1 and INF2 for influent, SLD1 and SLD2 for sludge, and EFF1 and EFF2 for effluent. The Charguia wastewater treatment plant (WWTP) treats mixed urban wastewater primarily from domestic, industrial, and hospital sources across the greater Tunis area. After transport to the laboratory, the influent and sludge samples, which contain high levels of suspended solids, were centrifuged at 13,000 rpm for 15 min at a temperature of 4 °C, resulting in the recovery of the microbial biomass in the pellet. However, the effluent samples, which are typically clearer and contain fewer solids, were filtered with a 0.22 µm diameter filter to concentrate the microbial content for DNA extraction. The AllPrep PowerViral DNA/RNA Kit (Qiagen, Hilden, Germany) was used for DNA extraction.

### Metagenomic sequencing

Metagenomic sequencing was performed on the Illumina Novaseq 6000 platform with $2 \times 150$ bp paired ends, enabling the precise identification and classification of bacterial communities present in each sample as well as the profiling of antibiotic resistance genes (ARGs), heavy metal resistance genes (MRGs), and MGEs. This thorough methodology was developed to study resistome-mobilome dynamics and understand their influence on resistance transfer. Raw sequences were submitted to the Sequence Read Archive (SRA) database of the National Center for Biotechnology Information under the accession number PRJNA1199733.

### Taxonomic profiling in Charguia WWTP

First, the metagenomic data were subjected to quality screening using Fast QC. To ensure high-quality sequencing, the Biobakery suite's built-in tool, Kneaddata v0.7.7-alpha (*McIver et al., 2018*), specifically designed for Illumina sequencing and integrating Trimmomatic and Bowtie2, was used for quality filtering. This process included trimming Illumina adapters and removing host contamination using default parameters, discarding reads smaller than 75 bp and with a Phred quality score below Q30. Metaphlan 4 v4.1.0 (*Manghi et al., 2023*) was used to profile microbial community composition at the species level. It uses a marker-gene-based approach for accurate species identification while minimizing false positives, making it ideal for profiling dominant taxa in wastewater microbiomes.

The retained sequences were evaluated by calculating richness and alpha diversity indices (Shannon, Simpson, and Phylogenetic Diversity Faith (PD_faith)) using Phyloseq v1.38 and Abdiv v0.2. Each sample was subjected to taxonomic profiling and the results were presented in bar graphs using R packages such as Phyloseq v1.38 and Microbiome v1.16. To identify bacterial taxa with notable differences in abundance between samples, the DESeq2 package v1.36 was used (*Love, Huber & Anders, 2014*). This method works by modeling count data with a negative binomial distribution, which helps account for variability in the data. It also adjusts for factors like library size and sequencing depth to ensure accurate

comparisons of taxa abundance. The results were visualized through a heatmap created with Complex Heatmap v2.16.0.

## Screening of ARG/MRG and MGE profiles

After preprocessing the metagenome sequencing data, the obtained clean reads were assembled de novo using MEGAHIT v1.2.9 (*Li et al., 2015*), a computationally efficient assembler optimized for short-read, large-scale metagenomic datasets. MEGAHIT is particularly suited for reconstructing contigs from complex microbial communities, making it ideal for wastewater metagenomic studies. The quality assembly report was then generated using QUAST v5.2.0 with a metagenomic-adapted pipeline (*Mikheenko, Saveliev & Gurevich, 2016*). A detailed overview of sequence read metrics is presented in Table S1. The annotation process began with a comprehensive approach using Prokka v1.14.5 (*Seemann, 2014*) with metagenome parameters, which is known for its comprehensive annotation capabilities. This was followed by more targeted identification of ARGs and MGEs using ABRicate v1.0.0 (*Seemann, 2020*). The results of both processes are compared and refined to ensure a coherent set of resistance profiles and a comprehensive understanding of resistance transmission within the ecosystem. A custom MGE database was created for ABRicate, complementing the analyzes with a curated collection of pretreated sequences from various sources, ultimately comprising 278 genes.

After performing data analysis, contigs with ARGs/MRGs and those with MGEs were extracted. These contigs were then subjected to in-depth analysis using Kraken v2.1.3 (*Wood & Salzberg, 2014*), a k-mer-based taxonomic classification tool that assigns reads to known genomes with high speed and accuracy. Kraken was used to cross-validate the taxonomic assignments previously obtained with Metaphlan 4, ensuring a more comprehensive and reliable identification of bacterial hosts. Furthermore, the metagenomic workflow optimized for Salmon v1.10.1 was used to accurately estimate the abundance of ARGs, which were expressed as the number per 1 million sequences in the metagenomic dataset. Using the quasi-mapping approach, the predicted protein-coding regions annotated with Prokka served as a reference for the quantification of genes. The *quant* command was then used to map reads against the indexed reference. Abundance was normalized by dividing raw ARG read counts by the total mapped reads and multiplying by one million to adjust for sequencing depth. To perform a complete analysis, the identified ARGs and MRGs were divided into classes based on the antibiotic and heavy metal to which they are resistant. These categorized genes were then visualized through bar graphs to provide a comprehensive representation. Furthermore, detailed heat maps were constructed separately for ARGs and MRGs to analyze the flux dynamics of resistance genes at different locations of the Charguia WWTP.

## Statistical analysis

The Analysis of Variance (ANOVA) test was performed to evaluate the significance of the richness and alpha diversity indices at the different lines within the Charguia WWTP. For the analysis of differential abundance, whether in taxonomic analysis or ARGs/MRGs analysis, the Wald test was used. To evaluate the co-occurrence of ARGs/MRGs and MGEs, a Pearson correlation matrix was also created and visualized as a heatmap.

## RESULTS

### Analysis of bacterial abundance and diversity across different sections of the wastewater treatment plant

Metagenomic approaches provide deep insights into bacterial composition and diversity, unlocking a thorough understanding of complex ecosystems. In this study, metagenomic sequencing generated an average of approximately 16.43 million paired-end reads, corresponding to 4.93 Gb of raw data and resulting in 2.29 GB of FASTQ files. After trimming and filtering, 4.27 Gb of high-quality reads were retained and subsequently assembled into contigs. Detailed sequencing information for the Illumina platforms used in this study is provided in Table S1. Taxonomic assignment identified 513 taxa for contigs. Figure S1 shows that the content, as indicated by the observed taxa, decreased in the wastewater samples after treatment. The results of the Shannon and Simpson diversity indices at different lines showed that certain taxa can dominate, which is particularly evident in the sludge samples; however, these differences were not statistically significant (Shannon; $F = 0.89$, $p$-value > 0.05, Simpson; $F = 0.302$, $p$-value > 0.05) (Fig. S1). A comprehensive bacterial composition study was conducted to compare bacterial community profiles in the Charguia wastewater treatment plant. The analysis revealed that *Proteobacteria* was the dominant phylum in all samples, accounting for 85.48% of metagenomic readings, particularly in wastewater samples. Bacteroidota were more abundant in sludge compared to the effluent, with relative abundances of 10.35% and 5.84%, respectively. Conversely, Firmicutes showed a notable decrease from 6.7% in sludge to 1.85% in the effluent. In contrast, Actinobacteriota exhibited an increase, with average abundances rising from 1.56% in sludge to 3.64% in the effluent. The phyla Fusobacteriota and Verrucomicrobiota were minimally represented in the effluent, accounting for only 0.03% of the microbial community (Fig. 1).

When analyzing the bacterial taxonomic profiles using metagenomic data, we observed apparent differences in the relative abundance of taxa between individual samples ($p$-value = 0.007). However, at the species level, no statistically supported differences in abundance were detected between origins ($p$-value > 0.05). These observations suggest that despite sample-specific variations, the overall microbial community structure may be influenced more by sample origin. Therefore, to improve the robustness and interpretability of the metagenomic analysis, samples were merged based on their respective origins.

At a more specific level, contigs-based analysis identified 29 taxa with notable differences in abundance across all samples, as shown in Fig. 2. Among these species, *Thiothrix eikelbomii* was particularly abundant in wastewater samples after treatment, accounting for 30.65%. Additionally, the examination of the metagenomic data revealed a wide range of species, including three *Pseudomonas* species (*P. psychrophila, P. fragi and P. fluvialis*), as well as *Psychrobacter* species and *Aliarcobacter cryaerophilus*, which were abundant in the influent but declined in the effluent. In the sludge, species such as *P. veronii, P. lundensis* and *P. weihenstephanensis*, as well as others identified as GGB46571_SGB64436 and GGB64019_SGB86406, were found to be declining in the effluent. As not previously mentioned, other *Pseudomonas* species were detected at varying abundances throughout the

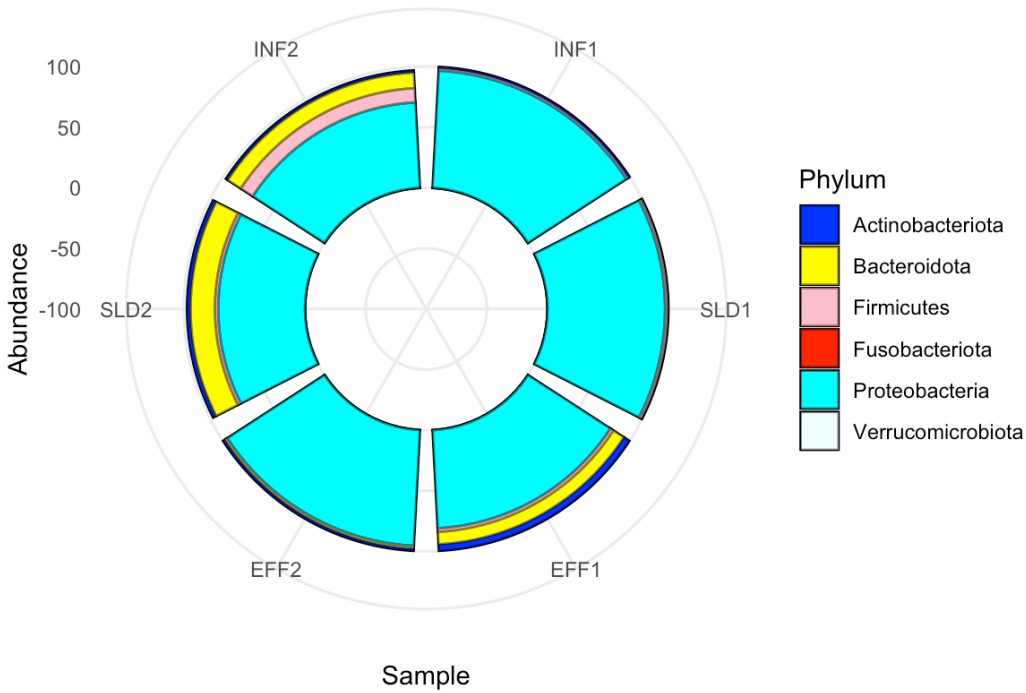

**Figure 1** **Phylum-level taxonomic composition across all samples from Charguia WWTP.** Circular barplots based on metagenomic data from the first (2022) and the second (2023) year of sampling.

treatment process (Fig. 2). These findings highlight the depth of metagenomic sequencing in characterizing microbial communities and their dynamic shifts within the wastewater treatment system.

## Abundance and diversity of ARG in the different lines of Charguia WWTP

After completing the annotation processes, a total of 46,187 reads were assigned to 42 different ARGs. These ARGs were present to varying extents in the influent (34 ARGs), sludge (39 ARGs), and effluent (38 ARGs) of the WWTP. Figure 3A shows that these identified ARGs belonged to 10 different antibiotic classes. The influent samples had the highest frequency of multi-resistance genes at 453,013 parts per million (ppm), with a significant reduction of 22% in the effluent. In addition, resistance to fluoroquinolone, fosmidomycin, tetracycline, peptide, and anthracycline were found in all samples, with average frequencies ranging from 7,801 ppm to 43,493 ppm. These genes either persisted in the effluent or became more prominent. The Penam class had the lowest abundance of ARGs with a value of 863 ppm. It was present in the sludge and remained in the effluent.

An in-depth study of the different ARG subtypes revealed that *mdt*A had the highest concentration in the sludge, averaging 80,027 ppm. However, this concentration in the effluent decreased to 58,370 ppm (Fig. 3B). Similar patterns were observed for other multiresistance genes such as *lmr*A, *mar*A, *bmr*A, *mex*A, and *yhel*. These genes collectively accounted for 20,343 ppm in inflow and experienced a significant decrease of 74%. On

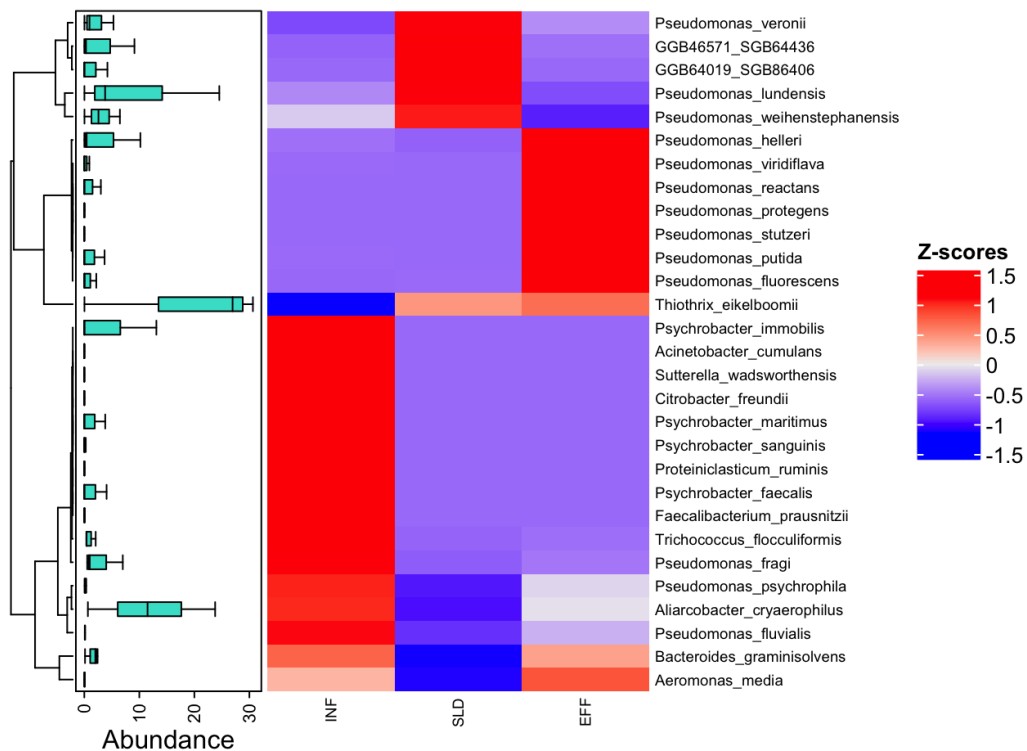

**Figure 2** Normalized abundance boxplots (*Z*-score) and heatmap of the differential abundance of the 29 identified metagenomic ASVs between the different sites (INF, SLD and EFF). Only significant values (*p-value* < 0.05) were retained.

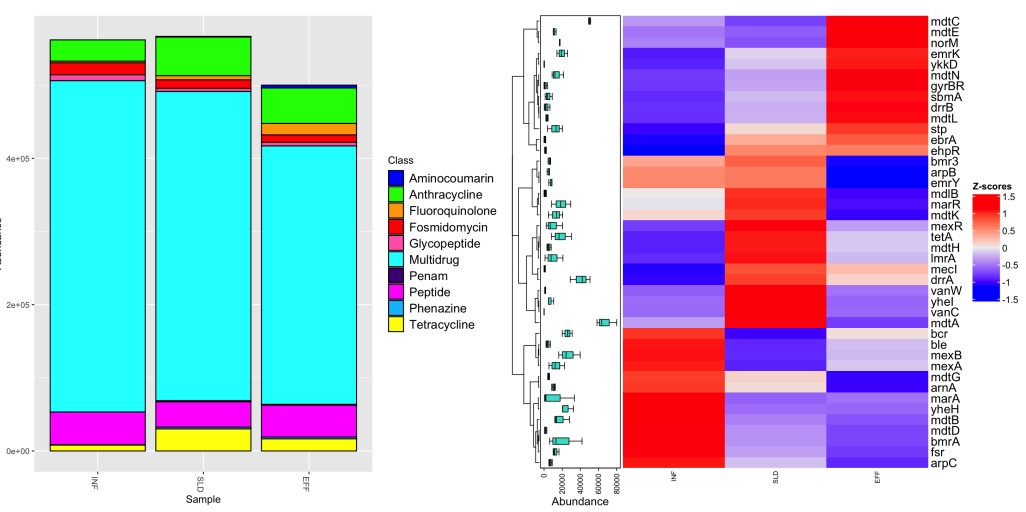

**Figure 3** Diversity of antibiotic classes (A) and heatmap illustrating differential analysis of ppm values for identified ARGs across various sites (INF, SLD and EFF) of Charguia WWTP (B).

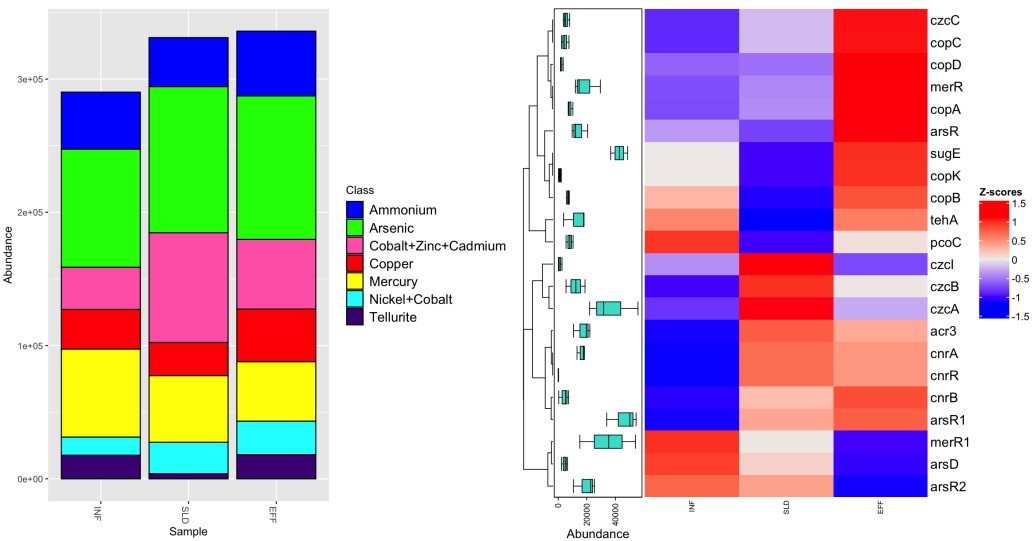

**Figure 4** Diversity of heavy metal classes (A) and heatmap illustrating differential analysis of ppm values for identified MRGs across various sites (INF, SLD and EFF) of Charguia WWTP (B).

the other hand, specific genes associated with resistance to glycopeptides (*van*W and *van*C), tetracycline (*tet*A), and anthracycline (*drr*A) showed lower levels in the effluent, ranging from 0 ppm to 42,050 ppm. Interestingly, certain genes such as *ykk*D, *ebr*A, *emr*K (multidrug), and *gyr*BR (aminocoumarin) were not detected in the influents but were present in the sludge and effluent and contributed to approximately 8,008 ppm after treatment. Genes such as *nor*M, *ddr*B, and *sbm*A, which confer resistance to multidrug, anthracycline and peptide, respectively, showed an increase of about 78% in effluent.

## Abundance and diversity of MRG in the different lines of Charguia WWTP

Regarding MRGs, a total of 16,247 reads were mapped to 22 MRGs, which were categorized into seven classes. Among these classes, two, namely nickel-cobalt and cobalt-zinc-cadmium, offered resistance to multiple metals (Fig. 4A). Sixteen MRGs were identified in the influent, while there were 20 and 21 MRGs in the sludge and effluent, respectively. After the treatment process, the most common class, arsenic resistance, showed a slight decrease in frequency in effluent from 109,704 ppm to 107,700 ppm. A similar trend was observed for the second most common class, genes resistant to cobalt-zinc-cadmium, with frequencies between 82,376 ppm and 52,346 ppm. The number of genes conferring resistance to mercury fell by 11%. However, the effluent had higher levels of ammonium, copper and nickel-cobalt resistance genes, accounting for approximately 48,580 ppm, 39,441 ppm and 25,173 ppm, respectively. Among all genes, tellurite resistance genes had the lowest abundance in effluent with an average frequency of 18,068 ppm.

Different subtypes of MRG were identified, as shown in Fig. 4B. The arsenic resistance class is represented by the *ars*R and *ars*R1 genes, which had average concentrations of 20,574 ppm and 54,521 ppm in effluent, respectively. Additional resistance genes were

observed in the effluent samples, namely *acr*3 and *ars*R2 for arsenic, *pco*C for copper and *mer*R1 for mercury. There was a 37% decrease in these genes. On the other hand, copper resistance genes including *cop*K, *cop*B and *cop*C showed a significant increase of 72% from sludge to effluent. Likewise, mercury (*mer*R) and nickel-cobalt (*cnr*B) resistance genes showed similar patterns with concentrations of 29,545 ppm and 7,282 ppm, respectively. Regarding cobalt-zinc-cadmium resistance genes, the effluent had higher concentrations of *czc*C (3,594 ppm) compared to genes *czc*A, *czc*B, and *czc*I, whose concentration decreased (55,807 ppm, 18,766 ppm and 2,676 ppm, respectively).

## Correlation between ARGs and MRGs in the lines of Charguia WWTP

The correlation matrix shows the selection of ARGs in the Charguia sewage treatment plant that were affected by heavy metal contamination (Fig. S2). The analysis initially focused on the ARGs that showed a decline in effluent and revealed various relationships. The *mdt*A gene, a common multidrug subtype in all samples, showed a positive correlation with *pco*C and *ars*R2, the genes associated with copper and arsenic resistance. On the other hand, a negative correlation was observed between cobalt-zinc-cadmium (*czc*B) and arsenic (*acr*3) resistance genes. In contrast, some ARGs such as *yhel*, *van*W, *tet*A and *drr*A, which are responsible for multidrug, glycopeptide, tetracycline and anthracycline resistance, showed an opposite pattern with a decrease in effluent samples. Furthermore, a connection between prominent ARGs in the effluent and MRGs was found. The results showed that aminocoumarin (*gyr*BR), anthracycline (*drr*B), and peptide (*sbm*A) subtypes were positively correlated with *mer*R, *cop*A, *cop*D, *cop*C, and *czc*C and conferred resistance to mercury, copper, and cobalt-zinc-cadmium, respectively. The *mer*R1 gene (mercury) showed a negative correlation with the ARGs. Interestingly, multidrug-resistant genes showed significant correlations, with genes such as *ebr*A and *emr*K being associated with the presence of cobalt-zinc-cadmium (*czc*B and *czc*A) and arsenic (*acr*3). However, *nor*M and *ykk*D showed a negative association with the arsenic resistance gene *ars*R1.

## Bacterial host association of ARG/MRG-carrying contigs

To identify the bacterial species that have ARG/MRG genes, we used Circos plots where links were created to visualize the relationships between ARG/MRG subtypes and microbial taxa, with the abundance within the samples providing the basis for the link weights (Fig. 5).

A comprehensive classification of 129 hosts was achieved, accounting for 56% of the resistance genes identified. Within the potential hosts (65 in the influent, 41 in the sludge and 72 in the effluent), the frequencies showed differences between samples. Certain ARGs were present in multiple hosts within the same sample. Notably, *Pseudomonas psychrophila* and *Pseudomonas ludensis* were specifically associated with resistance genes such as *mdt*A, *mar*R, *ars*R2, and *mer*R1, which were abundant in tributaries and sludge (Figs. 5A, 5B), but absent in effluents (Fig. 5C). This pattern was also observed for *mex*A, *nor*M, *ars*R2, and *mer*R1 in *Pseudomonas fragi* and *drr*A in *Aliarcobacter cryaerophilus* and *Psychrobacter sanguinis* (comparison Figs. 5A, 5B and 5C). *Acinetobacter johnsonii* carried the ARGs *van*W, *bmr*A and *emr*K as well as the heavy MRGs *cop*A, *czc*A, and *cnr*A in the effluent (Fig. 5A), which were eliminated after treatment as shown in Fig. 5C.

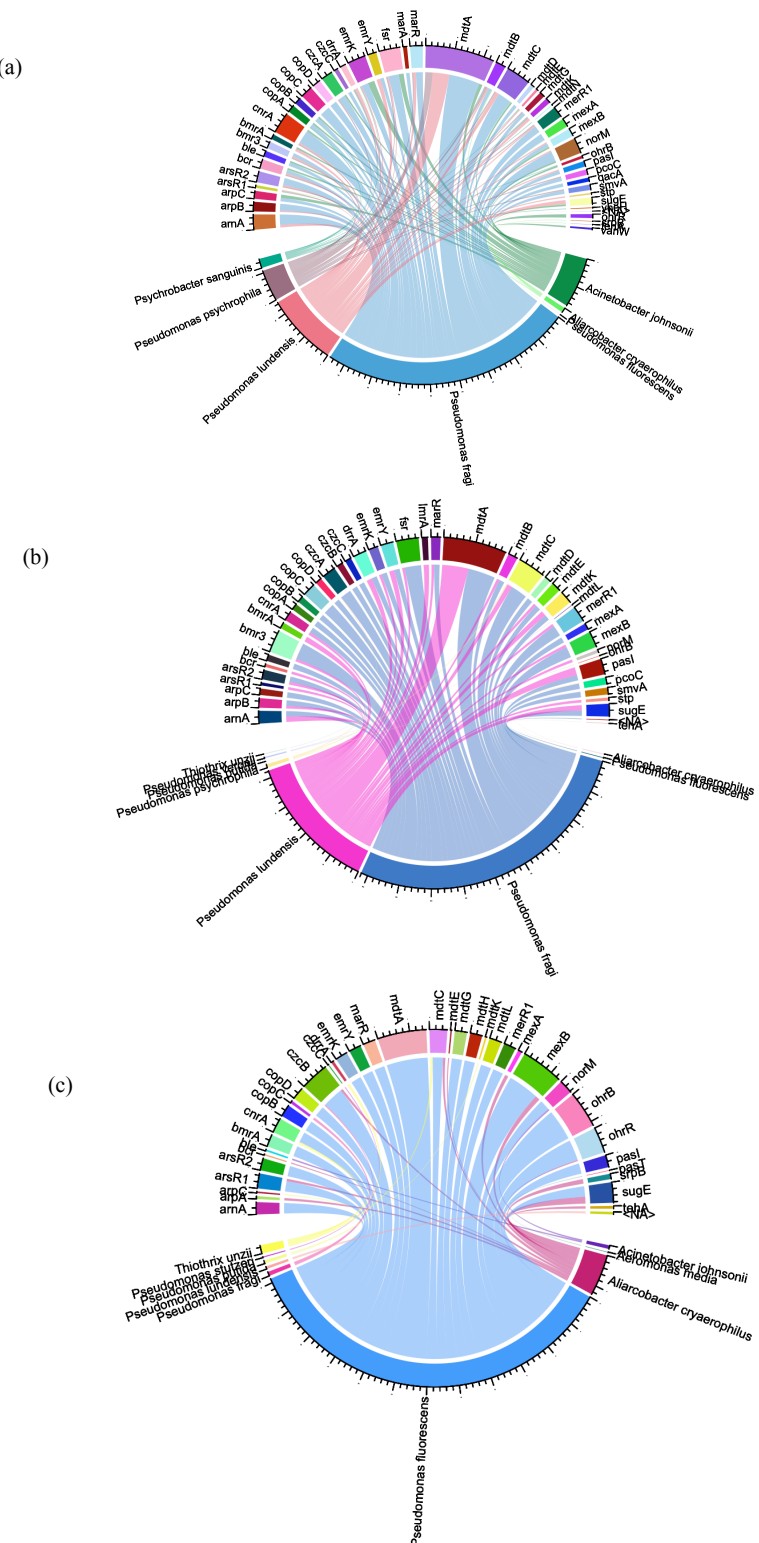

**Figure 5** Circos plots summarising the connection patterns between ARG/MRG subtypes and microbial taxa of the three sites of Charguia WWTP: influent (A), sludge (B), and effluent (C).

Conversely, there were persistent bacterial species in the effluent (Fig. 5C) that showed resistance, including *Aliarcobacter cryaerophilus* (*mex*B, *mdt*C, and *ars*R1), *Acinetobacter johnsonii* (*nor*M) and *Pseudomonas fragi* (*cop*C, *czc*B). Distinct resistance genes appeared in effluent from certain hosts, as shown in Fig. 5C, with *Thiothrix unzii* carrying *drr*A, *cnr*A, and *mdt*C. Other genes observed were *czc*B in *Aliarcobacter cryaerophilus* and *czc*C in *Pseudomonas stutzeri* in the effluent samples (Fig. 5C). Furthermore, Fig. 5C showed that the multi-resistance gene *mdt*K was exclusively present in *Pseudomonas putida* in effluent. *Pseudomonas fluorescens* was identified as the species with the highest number of resistance genes. These genes included *ars*R1/R2, *cnr*A, *mer*R1, *cop*A/B/D, and *czc*B (heavy metal resistance), as well as the ARGs *drr*A, *mex*B, *bmr*A, *mdt*A, *lmr*A, *emr*Y, and *mar*R (Fig. 5C).

## Gene transfer potential in the different lines of Charguia's WWTP

Various MGEs were discovered in all pipes of the Charguia wastewater treatment plant, which serve as potential transmission carriers. These elements included plasmids, conjugative transposons (Tn), and insertion sequences (IS). The heatmap in Fig. 6 shows the co-occurrence of ARGs/MRGs and MGEs, shedding light on their relationship and contributing to the understanding of bacterial dynamics in resistance gene transfer. Several plasmids such as IncFII, IncU, IncP(6), IncQ1, and IncQ2 were among the identified MGEs and played an important role in the spread of resistance genes. These plasmids showed different correlation profiles, with IncFII and IncU showing similar patterns of negative correlation with the multiresistance genes *mdt*A, *mex*R, and *lmr*A as well as the resistance genes arsenic *ars*R1 and mercury *mer*R. On the other hand, IncP(6), IncQ1, and IncQ2 showed inverse correlation profiles with both the aforementioned multidrug and heavy MRGs, except for *mer*R, which was positively correlated with IncP(6) and IncQ2.

In addition, IS26, IS91, and ISCR21, which share similarities with the IS91 family, were identified as IS elements. Elements associated with integron class I have also been detected, such as *int*I1 integrase genes. These elements and the integron were positively correlated with *bmr*A, *mar*R, and *mer*R1, while they showed negative correlations with *nor*M, *mex*R, and *cop*. The *ist*A and *ist*B genes, which are essential for the transposition process and associated with the IS21 family, showed an opposite pattern compared to the genes consistently.

Furthermore, the MGEs identified included transposons such as Tn916, which contained Orfs in addition to the *Int*-Tn916 integrase and *Xis*-Tn916 excisionase genes. Together, these components facilitated the conjugative transmission of transposons. Tn916 showed positive correlations with the multiresistance genes *mdtB* and *mexA* and the mercury genes *mer*R1, while it showed negative correlations with the arsenic and cobalt-zinc-cadmium genes *acr3* and *czcA*, respectively. These results highlight the significant association between resistance traits and transposons within the different lines of the wastewater treatment system.

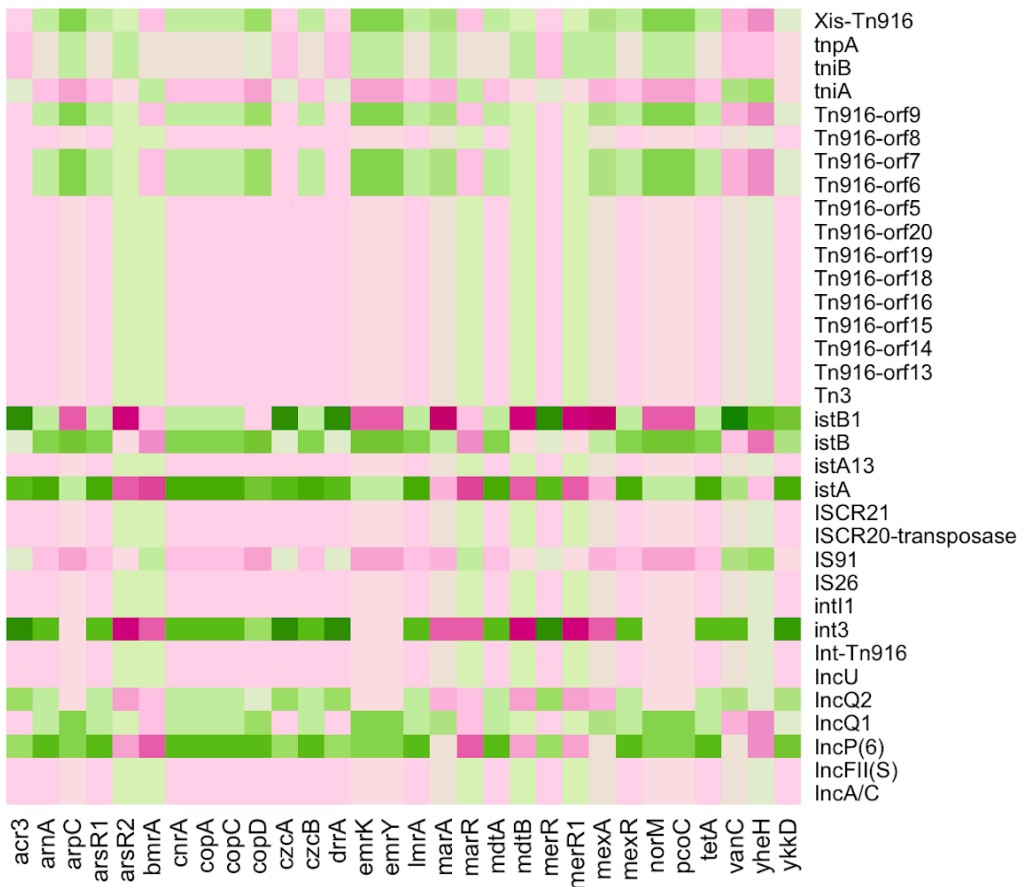

**Figure 6** **Heatmap illustrating the Pearson correlation between ARGs/MRGs (columns) and MGEs (rows) in the sampled Charguia WWTP.** Green = Positive correlation, Pink = Negative correlation.

## DISCUSSION

### Comparative analysis at the phylum level of the bacterial diversity

This study presents a preliminary investigation aimed at comprehensively monitoring antibiotic resistance and investigating the potential factors contributing to its spread within bacterial communities. Using a metagenomic approach, we achieved precise taxonomic identification at the species level, providing a comprehensive view of bacterial composition across influent, sludge, and effluent wastewater samples from the Charguia sewage treatment plant.

Our statistical analyses confirmed the dominance of Proteobacteria, along with other prevalent taxa, highlighting their significant role in the wastewater microbial community. Metagenomics also allowed for high-resolution characterization of dominant taxa, offering deeper insights into the functional and ecological dynamics of microbial populations within the treatment system. These findings from our study reinforce the value of metagenomic sequencing in capturing the complexity of wastewater microbiomes with high taxonomic precision (*Rieder et al., 2023*; *Srivathsan et al., 2015*).

In our study, *Proteobacteria* was the dominant phylum followed by *Bacteroidota*, which is consistent with results from studies conducted in WTTPs in Germany (*Numberger et al., 2019*) and Malaysia (*Yu, Tang & Lee, 2021*). Similarly, research in Tunisia on wastewater treatment plants in Kerkennah and Sidi Mansour revealed similar phylum abundances (*Boujelben et al., 2021*; *Mlaik et al., 2022*). These phyla are known for their important role in the removal of organic matter and nutrients, which explains their distribution in activated sludge (*Weissbrodt, Shani & Holliger, 2014*). In contrast to some studies that have found declines in these phyla in effluent (*Li et al., 2022a*), our study demonstrated their persistence. A study in Poland suggested that the inefficiency of the treatment process could be the reason for the persistent presence of these bacteria (*Niestepski et al., 2020*). While *Firmicutes* were initially abundant in the influent, *Actinobacteriota* were present in lower proportions. However, both persisted in the effluent, with *Actinobacteriota* showing an increase in abundance. This pattern has been observed in various WWTPs and their discharges to the environment suggesting a potential selection or enrichment process favoring *Actinobacteriota* during wastewater treatment (*Azli et al., 2022*; *Li et al., 2022b*; *Li et al., 2022c*; *Zieliński et al., 2022*). Their enrichment in effluents may reflect their key role in pollutant biodegradation as recently reported (*Zheng et al., 2024*), highlighting their potential contribution to the breakdown of organic contaminants and wastewater treatment efficiency.

## Species-level analysis of different resistant bacteria

A preliminary overview of the present genera was conducted to assess their distribution and dynamics before analyzing species-level variations. The metagenomic analysis revealed a diverse bacterial community, with key genera showing varying degrees of persistence and adaptation throughout the wastewater treatment process. Genera such as *Thiothrix* and *Pseudomonas* were among the most prevalent, highlighting their potential roles in both biodegradation and resistance mechanisms. *Thiothrix* genus are commonly found in WWTPs and not only lead to sludge bulking but also enhance granular sludge quality, sulfide oxidation, denitrification, and phosphate removal (*Gureeva et al., 2024*). Meanwhile, *Pseudomonas* genus is prevalent and is of significant concern due to their ability to develop antibiotic resistance (*Li et al., 2022b*; *Li et al., 2022c*). While some genera exhibited resilience and increased in abundance in effluents, others declined namely *Acinetobacter* and *Aliarcobacter*, likely due to treatment pressures. *Acinetobacter*, an opportunistic human pathogen, has been detected in different WWTPs as well as in surface waters and soil (*Zhang et al., 2020*). Likewise, *Aliarcobacter* genus is prevalent in WWTPs and can persist in effluents, posing potential health risks (*Kristensen et al., 2020*). These variations reflect the selective pressures exerted by wastewater treatment and underscore the importance of understanding bacterial dynamics at both the genus and species levels.

The analysis of species focused specifically on those that provide resistance. Within the *Proteobacteria* phylum, the study at the species level revealed *Thiothrix unzii, Pseudomonas putida, Pseudomonas stutzeri,* and *Pseudomonas fluorescens*, which were particularly common in the effluent samples. Recent research, including studies by *Spindler et al. (2012)* and *Li et al. (2022a)* has shown that *Pseudomonas* species such as *P. putida* and

*P. fluorescens* are opportunistic pathogens. These bacteria have been reported as potential carriers of multidrug resistance in soil and aquatic environments, contributing to the dissemination of antibiotic resistance under certain environmental conditions (*Camiade et al., 2020*; *Meng et al., 2022*). On the other hand, despite its pathogenic potential in humans, *P. stutzeri* has been classified as a heavy metal-resistant bacterium, and certain strains have shown promise for bioremediation in metal-contaminated environments, as reported by *Coelho da Costa Waite et al. (2020)*, *El-Bestawy & Aburokba (2017)* and *Kuroda et al. (2011)*. *Thiothrix* species were commonly found in sewage treatment plants and hot springs, as *Chernitsyna et al. (2024)* reported and *De Graaff, Van Loosdrecht & Pronk (2020)*. Furthermore, *Thiothrix* have been observed to be the most active bacteria in wastewater treatment systems and have a growth advantage over other bacteria in sludge, as *Brigmon, Furlong & Whitman (2003)* determined, *De Graaff, Van Loosdrecht & Pronk (2020)* and *Meng et al. (2022)*.

On the other hand, there was a decrease in the occurrence of several *Proteobacteria* species in effluent. These species include *Pseudomonas psychrophila, Pseudomonas fragi, Pseudomonas veronii, Pseudomonas ludensis, Psychrobacter sanguinis, Acinetobacter johnsonii,* and *Streptococcus parasuis.* This finding is consistent with previous studies by *Kosheleva et al. (2021)* and *Oliveira et al. (2021)* on multi-resistant bacteria. *Psychrobacter sanguinis* is typically found in low temperature marine environments and has been recognized as an opportunistic human pathogen (*Bakermans, 2018*; *Maruyama et al., 2000*; *Le Guern et al., 2014*). *Shi et al. (2023)* reported their presence only in the sewage treatment plant in China. In addition, *Acinetobacter baumanii* was the most commonly detected *Acinetobacter* species in effluent, as shown by various research studies (*Murphy et al., 2021*; *Zhang et al., 2009*). In our study, we found *Acinetobacter johnsonii* in effluent, suggesting that they can adapt to the treatment process by developing resistance (*Jia et al., 2022*; *Kisková et al., 2023*; *Murphy et al., 2021*). Despite the treatment, the pathogenic *Aliarcobacter cryaerophilus* from the strain *Campylobacterota* remained in the mud; its occurrence in the drains decreased, but was not completely eradicated. Similar results were reported by *Yuan et al.(2021)* reported and highlight their classification as resistant pathogens that can evade treatment (*Müller et al., 2020*; *Yuan et al., 2021*). Contrary to recent studies showing increased abundance in effluent (*Nataraj et al., 2024*), *Streptococcus parasuis*, a zoonotic pathogen within the *Firmicutes* phylum, was found primarily in effluent (*Ma et al., 2024*; *Qi et al., 2023*).

## Examination of the selective forces and co-selection mechanisms of antibiotic and heavy MRGs

Our research focused on the specific species identified as carriers of ARGs and/or MRGs in different lineages within the Charguia WWTP. These organisms, present either in the influent or in the sludge, were subjected to selective pressure after treatment. This phenomenon often led to a reduction in certain resistance genes until the water reached the effluent stage. However, we saw another trend where resistance genes increased in effluent. To understand these dynamics within bacterial communities, we first examined the resistance profiles and acquisition of genes conferring resistance to antibiotics and

heavy metals. Interestingly, our results showed that despite a decrease in abundance in effluent, *Pseudomonas psychrophila* and *Pseudomonas ludensis* had a significant presence of multi-resistance genes such as *mdt*A and *mar*R, as well as the genes *ars*R2 and *mer*R1, which confer resistance to arsenic and mercury, respectively. The possible co-selection mechanism is evident from the positive correlation between these resistance genes and provides information about their co-elimination after treatment. The situation is similar with the study by *Gibson et al. (2023)* also identified multiresistance genes that exhibit the same pattern. Furthermore, the presence of the *arp*C multi-resistance gene in effluent carried by *Pseudomonas psychrophila* adds an interesting piece of research and highlights the dynamic nature of resistance gene profiles during the treatment process.

According to *Kosheleva et al. (2021)*, *Pseudomonas ludensis* was reported to be a host carrying the *tet* gene. Other studies on heavy metals have examined arsenic and mercury resistance genes in wastewater treatment plants, although with different hosts such as *Klebsiella pneumoniae* and *Acinetobacter baumannii* (*Zagui et al., 2021*). *Ars*R was found to be most abundant in water samples of *Escherichia coli* (*Tseng et al., 2023*) and *Staphylococcus aureus* in macaques (*Monecke et al., 2022*). The influent and effluent samples had the same MRGs *ars*R2 and *mer*R1 as well as the multidrug genes *nor*M and *mex*A carried by *Pseudomonas fragi*. Interestingly, there was a positive correlation between *nor*M and *mer*R1, suggesting a possible cooperative relationship between these resistance mechanisms. In a study by *Finton et al. (2020)*, *nor*M, *mex*A, and *mer*R1 were also detected in *Stenotrophomonas maltophilia* from a Norwegian pond. Furthermore, *nor*M has been identified in hospital settings, particularly in *Staphylococcus aureus*, as reported by *Correia de Oliveira et al. (2023)*. In addition, a number of genes, including *mdt*A, *emr*Y (multidrug), *cop*B, *cop*C, and *cop*D (copper), were abundant in *Pseudomonas fragi*-associated effluent. These genes showed a positive correlation, suggesting a possible common mechanism or genetic link in response to selection pressure in the wastewater treatment process. Copper, commonly used as an antimicrobial agent in areas such as agriculture and industry, may play a role in this connection (*Arendsen, Thakar & Sultan, 2019*; *Coelho da Costa Waite et al., 2020*). The positive correlation between copper resistance genes and multi-resistance genes could indicate co-selection, where exposure to copper inadvertently favors the maintenance or co-occurrence of certain multi-resistance genes (*Baker-Austin et al., 2006*; *Zhang et al., 2019*).

The *drr*A genes carried by *Aliarcobacter cryaerophilus* and *Psychrobacter sanguinis* were detected in tributaries and sludge but were successfully eliminated after treatment. Our study is consistent with previous research on micropollutants in wastewater, such as the identification of anthracycline products in hospital wastewater (*Ajala et al., 2022*; *Ghafuria et al., 2018*) and the effective removal of these drugs through treatment (*Sharma et al., 2013*). Interestingly, *drr*A genes were found to confer resistance to doxorubicin in E. *coli* (*Kaur, 1997*) and to daunorubicin in *Streptomyces peucetius* (*Castillo Arteaga et al., 2022*; *Karuppasamy et al., 2015*), which differs from the identified host species differs in our study. In addition, *Thiothrix unzii*, was found to carry anthracycline resistance genes *drr*A in effluent. Our research also revealed the persistence of *Aliarcobacter cryaerophilus* in effluent, harboring the genes *mex*B, *mdt*A and *ars*R1 as well as an emerging gene *czc*B. Interestingly,

the MRGs *ars*R1 and *czc*B showed a negative correlation with anthracycline resistance genes, suggesting a complex interplay of resistance mechanisms and the possibility of co-selection (*Wang et al., 2020*; *Yang et al., 2020*). The results of *Acinetobacter johnsonii* in effluent reveal a diverse and complex collection of resistance genes, indicating the adaptability and versatility of the strain in response to environmental challenges. Various multi-resistance genes such as *bmr*A, *emr*K, and glycopeptide *van*W genes were detected. Previous studies have shown that *Acinetobacter johnsonii* also carries various resistance genes, including those that provide resistance to carbapenem, which was discovered in Chinese effluent (*Zong & Zhang, 2013*), as well as aminoglycoside, tetracycline and sulfonamide (*Castillo-Ramírez et al., 2020*). Our results also showed that the MRGs *cop*A/R, *czc*A, and *cnr*A are associated with *Acinetobacter johnsonii*. *Cnr*A genes were found in *Thiothrix unzii* persisting in effluent. These genes have been observed to persist in wastewater discharges and lead to contamination of the aquatic environment (*Yu et al., 2022*; *Zieliński et al., 2021*). Furthermore, the persistent presence of the *nor*M gene associated with *Acinetobacter johnsonii* in wastewater samples after treatment highlights its resilience and possible role in multidrug resistance. The complex correlation patterns between these genes suggest complicated interactions and selection pressures and highlight the need for further investigation into the mechanisms shaping the resistome of *Acinetobacter johnsonii* in wastewater treatment processes (*Timková et al., 2023*).

## Analyzing the mobilome: investigating the interactions between antibiotic and MRGs in Charguia WWTP

Our investigation of the ARGs-MRGs-host triad included analysis of MGEs in the wastewater treatment plant, which helped in the horizontal transfer of resistance genes between different bacterial strains. Activated sludge is considered the conductive medium for this transmission. The detection of resistance genes such as *drr*A, *cnr*A, *mer*R1, *ars*R and *czc*A in various bacterial species, including *Pseudomonas psychrophila*, *Pseudomonas fragi*, and *Pseudomonas ludensis*, indicates the horizontal transmission of these genes or co-resistance to antibiotics and heavy metals on their persistence in activated sludge. The anthracycline resistance gene *drr*A was consistently found in both influent and sludge samples associated with *Pseudomonas psychrophila* and *Pseudomonas ludensis*. Interestingly, these genes were present in *Pseudomonas fragi* within the sludge and *Pseudomonas fluorescens* and *Thiothrix unzii* in the effluent. The *drr*A gene in different bacterial species at different stages of wastewater treatment suggests horizontal gene transfer. The coexistence of anthracycline resistance genes with other severe MRGs suggests significant correlations that could explain their persistence in effluent. Some studies have suggested a self-resistance mechanism in certain strains to anthracycline, which could explain these genes in the discharges.

The importance of comprehensive analysis of the mobilome to track the transmission of ARGs and MRGs between bacteria within the wastewater treatment plant ecosystem is highlighted when assessing resistance profiles along with their associated hosts. Our research revealed multi-resistance genes in sludge, with *mdt*A and *mdt*K found in *Pseudomonas fluorescens* and *Pseudomonas putida* in effluent and *nor*M in *Acinetobacter johnsonii*. The

transfer of these resistance genes could be facilitated by plasmids such as IncU and IncQ1 and IS elements such as IS21, which were positively correlated in our study. These results are consistent with previous studies showing a high prevalence of such plasmids in wastewater systems containing multidrug-resistant strains of the *Pseudomonas*, *Escherichia* and *Sphingobacterium* genera. Furthermore, IncU plasmids, which are typically rare in *Enterobacteriaceae*, were identified as carriers of multidrug resistance, consistent with our results showing a positive correlation with the *yhel*, *mdt*B, and *mar*R genes. IncP plasmids were also found to co-occur to a significant extent with *mdt*A, *mex*R, and *lmr*A, particularly in *Pseudomonas* spp. contributes to multi-resistance. isolated from sewage treatment plant wastewater and surface water. Multi-resistance genes encoded in plasmids can improve the survival of hosts in discharges and provide a route for the introduction of additional resistance genes.

The IncF plasmid, known for its ability to carry multidrug-resistant genes in wastewater treatment plants (*Szczepanowski et al., 2005*; *Szczepanowski et al., 2004*; *Wibberg et al., 2013*), was discovered to contain mercury and arsenic resistance genes contained in bacterial isolates from hospital samples (*Evans et al., 2020*; *Rozwandowicz et al., 2018*). This confirms our previous results that IncFII showed a positive correlation with *mer*R1 and *ars*R2. Similarly, mercury resistance genes were detected in the pBP plasmid in *Pseudomonas stutzeri* (*Reniero et al., 1998*). These results shed light on the presence of *mer*R1 genes associated with *Pseudomonas fluorescens* in the Charguia sewage treatment plant. A study that examined the relationship between ARGs and IS elements using metagenomics revealed a high abundance of IS elements in wastewater treatment plants and water bodies (*Razavi et al., 2020*). Our results are consistent with this, as we identified various IS elements such as IS26, IS91, ISCR20, and ISCR21, which showed a positive correlation with *bmrA* and *marR*, which confers resistance to multiple antibiotics, and *mer*R1 for mercury resistance. Other studies also support the association between ARGs and class 1 IS elements and/or integrons, including intI1, which were abundant in a wastewater treatment plant in China (*Zheng et al., 2020*).

Our study is the first to use metagenomics to investigate resistance bacterial communities, resistome and mobilome profiles in a wastewater treatment plant in Tunisia. Despite the known potential of wastewater treatment plants, particularly activated sludge, to promote ARG proliferation among bacterial species, our results show a significant decline in resistance levels, particularly multidrug resistance. This indicates that the Charguia Wastewater Treatment Plant is likely effective in treating and eliminating certain resistant bacteria. We recommend further research using long-read technologies for high-resolution real-time monitoring to comprehensively analyze and explore MGEs and novel resistance elements. Our study highlights the need for additional research using long-read sequencing technologies and deeper short-read sequencing to improve species-level resolution and further enhance our understanding of the effectiveness of the Charguia Wastewater Treatment Plant in combating antibiotic resistance, ultimately contributing to the development of advanced and targeted wastewater treatment strategies with broader implications for global public health.

## ACKNOWLEDGEMENTS

We appreciate the intervention of all the staff involved in the present topic.

### Funding

The authors received no funding for this work.

### Competing Interests

The authors declare there are no competing interests.

### Author Contributions

- Chahnez Naccache conceived and designed the experiments, performed the experiments, analyzed the data, prepared figures and/or tables, authored or reviewed drafts of the article, and approved the final draft.
- Chourouk Ibrahim conceived and designed the experiments, authored or reviewed drafts of the article, and approved the final draft.
- Abdennaceur Hassen conceived and designed the experiments, authored or reviewed drafts of the article, and approved the final draft.
- Maha Mezghani Khemakhem conceived and designed the experiments, authored or reviewed drafts of the article, and approved the final draft.

### Data Availability

The data is available at NCBI: PRJNA1199733.

### Supplemental Information

Supplemental information for this article can be found online at http://dx.doi.org/10.7717/peerj.19682#supplemental-information.

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
