# Peer review of "Metagenomics-based analysis of mobile genetic elements and antibiotic/metal resistance genes carried by treated wastewater"

_PeerJ, doi:10.7717/peerj.19682_

## Round 0.1 · original submission · Major Revisions

Please address the concerns of the reviewers and pay special attention to the methodological and analytical improvements needed.

Since one of the two reviewers recommended rejection, I may have to call an additional one.

·

Basic reporting

The article submitted by Naccache et al, aims to define the bacterial community and resistome present in wastewater treatment plants in Tunisia. The writing is of a high standard, clear and formally presented. The literature review is robust and follows a sound logic introducing each topic in a structured manner. The figures and tables are clearly presented and support the text in a way one would expect.

Experimental design

The research question was clear but the experimental design is slightly clumsy, I’m unsure why both 16S rRNA and WGS metagenomics were both used. These data rarely align when compared, and their value in this type of investigation when metagenomics has been applied is superfluous. The results from the WGS with more robust analysis like coassembly and binning would have been better and answered the questions in a simpler manner, instead the authors seem to have replicated analysis unnecessarily, for example using Metaphlan, Megahit and Kraken2. There is also missing information in the methods on how the links in the Circos plots were generated and how the reference were made to allow Salmon to determine the abundance of ARG/MGE genes.

Validity of the findings

Determining species level taxonomy with the amount of data used and the truncated analysis pipeline is ambitious, if not a little careless. There are still limitations in assignment to a species level, most are lessened using more downstream pipelines, which is outlined by the authors of Kraken2, by using Braken to incorporate minimum k-mer counts per genome for increased confidence. In a field of research where 90-100 million reads per sample is gold standard and 20-40 million reads is classed as the accepted minimum, only using 6 million reads seems unfeasibly low. Especially in the context of some of the claims being made. It would be more prudent to stick to a genus level taxonomic assignment after further rigorous analysis. The assignment of the 16S rRNA to a species level should be avoided. The length of the V3-V4 region of the 16S is not able to differentiate between most species so genera is the accepted level of taxonomy used.

Additional comments

Data not openly available

Comment in abstract about the effectiveness of WWTP, unsure what this is referring to.

There is minimal discussion on the genera identified and if they were expected to be present.

Spelling of integrons on line 106

Double full stop on line 288

Missing in text references to tables for sequence read metrics in methods.

No need to refer to Abricate as a tool, just Abricate is fine line 183 and 186

Paragraph (lines 242-263) is a little confusing when referring to ASV’s from the WGS metagenomics, not sure if this is an error (I assume it is) but this needs clarifying and methods adding if this was done too.

Unsure how the part per million calculation was achieved in relation to ARG’s. The concentration of antimicrobials in the wastewater was not investigated and gene are not referred to in this manner.

Reviewer 2 ·

Basic reporting

In this study, the authors performed amplicon and metagenomic sequencing of 6 wastewater treatment plant samples to characterize the microbial communities, in particular the antibiotic resistance genes and metal resistance genes.

Experimental design

The experimental part in this study is mostly sampling, DNA extraction and sequencing. Overall it is well designed.

Validity of the findings

However, I find this work suffer from substantial flaw in data analysis, and the authors have to do a major overhaul to make the analysis more rigorous, robust and convincing. For example, the taxonomic composition from amplicon sequencing and metagenomic sequencing is very different. Although different methods can give slightly different results, such major discrepancy between methods for the same sample really suggests issues in how the data processing was carried out.
Furthermore, misuse/misunderstanding of basic biological concepts are common in the manuscript, and I recommend the authors to systematically understand these basics before repeating the analysis. For instance, Line 219 and several other sentences, ASVs are only for amplicon reads, not for contigs. It is inappropriate to assign ASVs for metagenomic contigs.
Finally, many conclusions are drawn based on statistical tests with very large p values, sometimes even higher than 0.1. The low level of statistical significance makes it hard to convey convincing conclusions.

---

## Round 0.2 · Major Revisions

Please address all reviewers' comments.

·

Basic reporting

I had no major concerns about the reporting in the first version. Its well written and well presented.

I still think the calculations are incorrect with the number of reads. They state an average of 2GB of data was generated, which is 6.6 million reads. Their supplementary data does state 16 million reads which would be 5GB of data based on 2x150bp reads.

Experimental design

The methods and results have been edited to be more understandable and make more sense. The authors have made some good points regarding some analysis choices.

I still think using ppm is misleading. It is standard practice to use tpm in transcriptomics but ppm is specifically used in chemical analysis.

Validity of the findings

I still think species level assignment is unfeasible with this amount of data.

Reviewer 3 ·

Basic reporting

• The overall redaction quality is not good, leading to a poor comprehension of the text and making results interpretation quite laborious. To mention some points:
- The use of the term “wastewater” is unclear, it appears to be used as a synonym of “effluent”. Considering the division between “influent” and “effluent” to be central to the manuscript, I think this is an important issue to address. This confusion makes the interpretation of results very difficult.
- It is unclear for example in lines 221-225 whether the comparison is always carried out between influent and sludge.
- Ideas are not clearly explained, requiring the reader to go through them multiple times (e.g.: Lines 72 – 74). There are some broken sentences, probably product of draft editing (e.g.: Lines 74 – 78).
- Line 39 is a little unclear, what do authors mean by “88 combinations of drug resistant bacteria”? Do they mean 88 different combinations of resistances?
- There is no articulation between some sections of the introduction, leading to abrupt changes in the read flow.
- Section 2.1 is titled: “Analysis of bacterial abundance and diversity at various wastewater treatment plants (WWTP)” but according to the methods section the samples were taken from different sections of the same WWTP.
- Overall, punctuation revisions are required. For example, in line 76 there is a stop after “plants” that should not be there)
• Authors mention many times that a p-value >0.05 corresponds to significance
• More detail on some aspects would be appreciated in order to help the reader assess the information authors are giving. For example:
- The differences in the locations mentioned in the text and what makes them stand out (e.g.: Monastir, Chotrana, Charguia, Sousse), are they heavily populated? Do they present different levels of industrial activity or pollution?
- Line 96-98: Do these heavy metals levels correlate also with higher resistance rates in the Bizerte Lagoon in comparison to less polluted lagoons?
- Line 254: Do authors refer to multidrug-efflux pumps when they say “multi-resistance genes”?
- Antimicrobial resistance in urban wastewater, wastewater treatment plants and even in treated wastewater is a phenomenon that occurs practically worldwide. I understand why authors chose to highlight studies carried out in Tunisia but I do not follow the reasoning behind listing Brazil as a single example. I believe a worldwide mention to the problem with a later focus on Tunisian findings would make the introduction more structured and senseful.
- The first two paragraphs present data about Africa from 2022 while the data on Tunisia is from 2015. Additionally, it feels as if Tunisia was responsible for the whole African epidemiology and that antimicrobial consumption in this country would explain the observed resistance in the whole continent. I suggest either leaving lines 43-46 out or adding a previous paragraph commenting on the resistance levels in Tunisia in comparison with the rest of the continent so that it backs-up the antibiotic consumption statement.
• Minor aspects:
- An explanation of the abbreviations even if commonly used would be beneficial for any reader to interpret the terms, an example of this can be “AMRs” in line 35 and “ARGs” in line 110.
- I would suggest keeping citations consistent, some parts of the text mention for example (Klein et al., 2018) while others read, for example, as “Marti et. al (2014) emphasized”
- Line 52: I’m uncertain about the statement that pollutants “lead to the development of resistance”.
- Figure S2 would benefit from a legend explaining what the gradient bar represents.
- Citations for statements like those in line 346 would be appreciated

Experimental design

My major concern about the experimental design is that the authors apply statistical tests to 6 samples, divided into 3 locations. That gives 2 technical replicates per location which are also technically not replicates, since the sampling efforts are one year apart. For whichever statistical comparison, 2 samples are not enough. In addition to this, based on composition similarities, the authors merge the replicates, leaving them with basically one sample per location.
Other experimental design observations/questions (in order of relevance):
• Why was the Charguia treatment plant selected? Does it represent an average urban location in Tunisia?
• Why did the authors select November as a sampling date. As is the case with any other wastewater treatment plant, the contents of the inflow and also outflow are subject to seasonal variations. I wonder if the authors would have seen different effects if sampling had taken place during a different season or multiple times throughout a year.
• Where did the sampling for the influent and effluent take place? Was it directly at the entrance/exit of the WWTP? Also, more information on the type of waste water the plant is treating would be highly beneficial, especially for result interpretation and discussion.
• Influent and wastewater samples are mentioned to be treated differently in lines 138-140. Why was this? – Clarification on the terms is needed.
• Kraken when used on metagenomics short read data or on short DNA fragments will usually not be able to classify them as they do not contain enough information to make a reliable call. I’m therefore wondering if the approach used by the authors to assign taxonomic identity to the ARG containing contigs is reliable. Could the authors please give details on the length of the ARG carrying contigs? Additionally, Kraken is dependent on the used database and the confidence parameter, I think both should be reported. On this note, also, the programmers of Kraken suggest themselves to run Bracken afterwards due to issues estimating taxonomic labels, specially at the genus/species level. (https://doi.org/10.7717/peerj-cs.104)
• Since taxonomy assignment is heavily database dependent, I would appreciate if authors could list the employed database. In my experience very few databases and algorithms are able to assign species and their abundance reliably, I would tend to think that analysing in such a deep level would return a lot of “Unknown” or “Unclassified” species. My recommendation would be to analyse data at the genus level, something that the most robust algorithms and databases can handle.

Knowing who is doing what in a metagenomic context is one of the most challenging things to do. Maybe quantifying ARGs and taxa (both from the reads) and then figure out correlation values between these variables would give a different take on this.

• If possible, sequencing data should be made available in some public repository (i.e.: GenBank, ENA)
• Information on the Q-cutoff for quality trimming would be appreciated
• A brief description of the statistical approach the authors used to identify representative bacterial taxa in different samples would be beneficial.
• An ANOVA approach was used, did the authors verify homoscedasticity? Did they apply a post hoc test?
• Line 254-257: How are ppm calculated? Are they reads per million. If so, I’m having trouble understanding how 46 187 ARG matching reads turn to 453 013 reads per million.
• Line 155: Do authors mean human DNA when they mention “host contamination”?

Validity of the findings

Regarding the validity of the findings, my main concern still lies on the statistical associations drawn from so very few samples.
Some other concerns (In order of relevance):
• Authors mention that the observed taxa decreased after treatment but the results in Figure S1 don’t support this claim. Boxplots for observed taxa, Shannon and Simpson indexes show notorious overlap between influent and effluent samples indicating that there might not be significant differences. To back this up, none of the reported p-values, assuming they derive from the ANOVA tests, are significant.
• Authors state that the variations observed using Pseudomonas sp., as an example, reflect the selective pressure applied by WWTPs. In spite of this, in section 2.1 – Line 241 they mention that their abundance is higher in the influent and decreases in the effluent.
• Is there an experimental confirmation for the statement that these organisms were subject to selective pressure after treatment (Line 463)?
• Authors discuss many potential reasons why MRGs and ARGs can be selected and co-selected based on available literature (e.g.: Copper from agriculture and industry, anthracycline from hospitals). In spite of this, they fail to mention the type of wastewater the Charguia treatment plant receives. I believe this is a key factor to analyse the reasons leading to the selection the authors claim to observe.
• Line 230: No mention is done regarding the parameters the authors are comparing to make this affirmation.
• Figure 2 is confusing, why is there only one boxplot per species while there are three potential sources (influent, effluent, sludge) ? Also, the heatmap represents Z-scores in each site for each species, making it difficult to assess abundance.
• It is difficult to judge because of the nature of the figures but genes like copA and emrK seem to have high Z-scores in the effluent, suggesting they are somehow above the mean. This contradicts the authors statements in lines 330 – 331 that say that these genes are eliminated after treatment.
• Line 541: It may suggest horizontal gene transfer but it does not confirm it is taking place at this WWTP.
• It is curious to see that every correlation (negative or positive) between ARGs/MRGs and MGEs are significant.
• I would appreciate some clarification on how the percentage of reduction was calculated and which were the criteria to determine significance of reduction.
• I wonder how the boxplots were produced for figure 2 if there are (at most) 2 samples per condition (influent, effluent, sludge). Figure 2 also mentions p > 0.05 as significant.
• Is the affirmation in lines 350-352 derived from this study? If it is, I would appreciate details on this association, if not, a reference should be added.
• Line 360: Was the whole class 1 integron queried or just intI1?
• The first paragraph of the discussion reads as if it was referring to another study.
• Lines 428-429 are an over generalisation

Additional comments

The manuscript “Metagenomics-based analysis of mobile genetic elements and antibiotic/metal resistance genes carried by treated wastewater” explores the bacterial community, antibiotic resistance genes, mobile genetic elements and metal resistance genes dynamics during the wastewater treatment process in a single wastewater treatment plant in Tunisia. The work attempts to gain insights of a highly important topic worldwide proposing relevant research questions. The authors employ current methodologies in order to deeper explore such a pressing issue.
Most of current studies employing metagenomics come from high income countries, the fact that this work is set in an LMIC is an added value since it has the potential to provide insights of operation conditions and a comparability point both for other LMICs, but also for high income countries.
In spite of this, there are several points that need to be brought to your attention.

---

## Round 0.3 · Minor Revisions

Please address the remaining reviewer comments.

Reviewer 3 ·

Basic reporting

I would like to thank the authors for taking the time to reply every comment and question raised on my side. I am mostly satisfied with the provided answers and the changes made to the manuscript. I also thank the authors for instructing me in the various uses of citation formatting.

Experimental design

As expressed in the initial review, my major concern is related to the amount of samples and replicates which are not enough to provide proper statistical support for significant trends/differences observed between samples. The authors express in their response letter that they are aware of the limitations of their sampling strategy and state that their work is mostly exploratory, therefore trends should be considered to have biological value without the strict need for significance. I believe this is a relevant aspect that should be made clear, the use of phrases like “significant differences” should be somehow modified in accordance to this.

Validity of the findings

The findings are valid as long as they are treated as descriptive data with trends and associations. Because of the experimental design, I don’t think making statistical affirmations is the right thing to do here.

Additional comments

After applying changes there are some minor format aspects that should be polished, for example:

Line 260 reads “plants” where it should be “plant”.

In line 502 the Ps should be in italics, the same with the P in line 509.

Authors mention that they already have uploaded the reads to a publicly accessible database and that the data will be released after publication. It is important for scientific transparency to provide the accession number in the manuscript itself.

---

## Round 0.4 · accepted · Accept

Thanks for addressing all the comments!

Reviewer 3 ·

Basic reporting

Authors have addressed all of my concerns and I appreciate it. I have no more comments.
I still found some typos left after editing (e.g.: Line 170, double dots) but that is, of course, very minor.

Experimental design

No comment

Validity of the findings

No comment